# In-situ vertical characteristics of optical properties and heating rates of aerosol over Beijing

Ping Tian[1], Dantong Liu[2*], Delong Zhao[3], Chenjie Yu[4], Quan Liu[1], Mengyu Huang[1], Zhaoze Deng[5], Liang Ran[5], Yunfei Wu[6], Shuo Ding[2], Kang Hu[2], Gang Zhao[7], Chunsheng Zhao[7], Deping Ding[1,3]

1. Beijing Key Laboratory of Cloud, Precipitation and Atmospheric Water Resources, Beijing, 100089, China.

2. Department of Atmospheric Science, School of Earth Sciences, Zhejiang University, Hangzhou, Zhejiang, 310027, China.

3. Field Experiment Base of Cloud and Precipitation Research in North China, China Meteorological Administration, Beijing, 100089, China.

4. Centre for Atmospheric Sciences, School of Earth and Environmental Sciences, University of Manchester, Manchester, M139PL, UK

5. Key Laboratory of Middle Atmosphere and Global Environment Observation, Institute of Atmospheric Physics, Chinese Academy of Sciences, Beijing, 100029, China.

6. CAS Key Laboratory of Regional Climate-Environment for Temperate East Asia, Institute of Atmospheric Physics, Chinese Academy of Sciences, Beijing, 100029, China.

7. Department of Atmospheric and Oceanic Sciences, Peking University, Beijing, 100871, China

*Corresponds to*: Dantong Liu (dantongliu@zju.edu.cn)

**Abstract.** Characterizing vertical profiles of aerosol optical properties is important because only relying on the surface or column-integrated measurements is unable to unambiguously constrain the radiative impacts of aerosol. This study presents series of vertical profiles of in-situ measured multi-wavelength optical properties of aerosols during three pollution events from Nov. to Dec. 2016 over the Beijing region. For all pollution events, the clean periods (CP) before pollution initialization showed a higher scattering Ångström exponent and a smaller asymmetry parameter ($g$), and with relatively uniform vertical structures. The heavy pollution (HP) periods showed increased particle size, causing these parameters to vary in the opposite way. During the transition periods (TP), regional transport of aged aerosols at higher altitude was found. The AERONET aerosol optical depth (AOD) matched the in-situ measurements within 10 %, however the AERONET absorption optical depth (AAOD) was 10-20 % higher than the in-situ measurements, and this positive discrepancy increased to 30 % at shorter wavelength. The absorption of brown carbon (BrC) was identified by the increased absorption Angström exponent (AAE), and the heating rate of black carbon (BC) and BrC was estimated by computing the spectral absorption coefficient and actinic flux calculated by a radiative transfer model. BC and BrC had a heating rate up to 0.18 K/h and 0.05 K/h in the planetary boundary layer (PBL) respectively during the pollution period. The fraction of BrC absorption increased from 12 % to 40 % in the PBL from CP to HP period. Notably, a higher contribution of BrC heating was found above the PBL under polluted condition. This study gives a full picture of shortwave heating impacts of carbonaceous aerosols during different stages of pollution events, and highlights the increased contribution of BrC absorption especially at higher altitude during pollution.

41

# 1 Introduction

The optical properties of aerosol, which is how aerosol scatters or absorbs solar radiation, have caused important radiative impacts on earth system (IPCC2013). The optical properties depend on the particle size (Bergin et al., 2001), refractive index (Ebert et al., 2002; Quinn, 2002) and mixing state of aerosols. There are still large uncertainties in evaluating the radiative forcing of aerosol especially in east Asia region due to lack of information on vertical distribution of these parameters (Liao and Seinfeld, 1998; Ramanathan et al., 2001; Li et al., 2017). Previous studies showed that the surface observation or column-integrated measurements may not provide sufficient information to derive vertical profiles of aerosol optical properties (Andrews et al., 2011; Rosati et al., 2016). Modelling studies find the radiative forcing impact is sensitive to the aerosol vertical distribution (Haywood et al., 1998), and especially for the absorbing aerosol for example black carbon (BC) will exert different climatic impacts depending on the location of the aerosol layer (Yu et al., 2002; Ban-Weiss et al., 2011; Wilcox et al., 2016). Though most aerosols are contained inside the planet boundary layer (PBL), the climatic sensitivity to absorbing aerosol rapidly increases with altitude (Ramanathan et al., 2001; Hodnebrog et al., 2014; Nazarenko et al., 2017). Absorbing aerosol above the PBL has the potential to suppress the PBL development and enhance the inversion cap at top of the PBL (Ding et al., 2016; Wang et al., 2018c), further aggravating the pollution. However, this impact depends on the location of the absorbing layer which may also promote the convection by heating the layer above (Koch and Del Genio, 2010; Yu et al., 2019). It is therefore important to characterize the vertical profile of absorbing component in the atmosphere in order to understand its influences on atmospheric thermodynamics.

The North China Plain (NCP) has raised great attention in the recent decade because of the severe air pollution and high frequency of hazy days over this region. The causes of pollution have been widely investigated through surface measurements (Zhang et al., 2013; Zhang et al., 2015; Zhong et al., 2018), however only limited studies have considered the evolution of pollutants in the vertical direction (Tian et al., 2019; Wang et al., 2018a). It was found that the surface aerosol concentration over Beijing not only depended on the emission but also the vertical structure of the aerosol distribution, which was largely dependent on local and synoptic meteorological conditions (Ran et al., 2016a; Zhao et al., 2019a), such

as the mountain chimney effect over the Beijing region which may introduce enhanced aerosol loading to high altitudes (Chen et al., 2009). The light-absorbing aerosol mainly includes the species of black carbon (Bond et al., 2013), brown carbon (Lack and Cappa, 2010) and dust (Klingmüller et al., 2019), which have different spectral sensitivities to solar radiation. Different aerosol components dominate at different environments, and the heating rates caused by various aerosol sources have been studied over the world, e.g. for the anthropogenic sources over north America (Gao et al., 2008; Sahu et al., 2012; Liu et al., 2015b), Europe (Ferrero et al., 2014; Ferrero et al., 2018) and south Asia (Chakrabarty et al., 2012; Shamjad et al., 2015), and biomass burning sources over north and south America (Saleh et al., 2015; Zhang et al., 2017). However, there is only sparse data regarding the vertical structures of heating rates. The calculation were performed for single species such as BC or BrC but most did not consider the co-impacts of all species (Chakrabarty et al., 2012; Chung et al., 2012; Shamjad et al., 2015). In the lower free troposphere, the interaction of aerosol-induced heating with boundary layer dynamics has raised much attention recently, as it may play important role in suppressing boundary layer development hereby exacerbating the local pollution (Li et al., 2017). The heating rate caused by light absorbing aerosol was reported to vary as a function of height and range at 0.3-2.1 K/day for the polluted PBL over Europe (Kedia et al., 2010; Ferrero et al., 2014; Ferrero et al., 2018), and 0.3-2.5 K/day for south Asia (Tripathi et al., 2007; Ramana et al., 2007; Ramachandran and Kedia, 2010; Chakrabarty et al., 2012). Only a limited number of reports are available for east Asia.

This study chose three typical pollution events occurring in wintertime over Beijing, and performed continuous flights on daily basis for each event. The vertical profiles of multi-wavelength aerosol optical properties were in-situ characterized, accounting for all stages during pollution events from the pollution start, full development and cease. The directly measured optical parameters were used as inputs for radiative transfer calculation, hereby the heating rates of light-absorbing aerosols, including black and brown carbon (BrC) were estimated. The results here provide a full picture of vertical profiles of aerosol optical properties over Beijing region and investigate the radiative forcing effect of aerosol during the heavy pollution events.

**2 Instrumentation and data analysis**

A Kingair 350ER turbo aircraft in Beijing weather modification office was employed for the in-situ measurements over Beijing during the 2016 winter in this study. Meteorological parameters including the temperature, relative humidity, pressure, wind direction and wind speed with a time resolution of 1 s were measured by the Aircraft Integrated Meteorological Measurement System (AIMMS-20, Aventech Research Inc, Canada), which was calibrated annually. The aerosol instrumentation inside the cabin was connected to an isokinetic inlet (Model:1200, Brechtel Inc, USA), which can deliver particle with a high transport efficiency (90%) for sub-micrometer particles. The room temperature (25 °C) in the cabin had a self-drying effect when the temperature inside was higher than outside the cabin, in addition to which, a silicate dryer was used of all instruments to maintain a sampling RH lower than 40%.

In-situ measurements of aerosol optical properties were performed during three pollution events over Beijing in Nov. 15th to Dec. 21th 2016, including 14 flights covering the start, development and cease stage for each pollution event. All flights were conducted around midday when the PBL was well developed. Table 1 summarizes the information of each flight. In order to compare the AOD from AERONET and to calculate the vertical heating rates, only the cloud-free vertical profiles are used. In this study, three flights (20161117 12:00, 20161117 15:00, 20161118 12:00) were observed with cumulus clouds (Table 1). The in-cloud data in this study was screened out according to the in-situ measured cloud number concentration and liquid water content. Data with a total number concentration of more than 10 cm$^{-3}$ and liquid water of more than 0.001 g m$^{-3}$ are not included in the following analysis (Deng et al., 2009). A micro pulse lidar (MPL, Sigma Inc, USA) was employed to measure the temporal evolution of aerosol extinction vertical profiles, and the vertical wind profile was measured by a wind profile radar with a vertical resolution of 150 m.

## 2.1 Aerosol optical properties

The aerosol scattering ($\sigma_{sca}$) and hemispheric backscattering ($\sigma_{bsca}$) coefficients at $\lambda$=450 nm, 525 nm, and 650 nm were measured by an integrating nephelometer (Aurora3000, Ecotech Inc, Australia), and the flowrate of Aurora3000 was maintained at 4 L/min during flight. The baseline of Aurora3000 in real time

was corrected for Rayleigh scattering of gas molecule at different air pressure (Fig. S1). In addition, the $\sigma_{sca}$ and $\sigma_{bsca}$ at all wavelengths were corrected for truncation affects (Anderson and Ogren, 1998; Müller et al., 2009).

The scattering Ångström exponent (*SAE*) measures the wavelength dependence of $\sigma_{sca}$ assuming a power relationship with λ, expressed as:

$$SAE = -\frac{\ln(\sigma_{\lambda1}/\sigma_{\lambda2})}{\ln(\lambda_1/\lambda_2)}, \tag{1}$$

where $\sigma_{\lambda1}$ denotes the $\sigma_{sca}$ at $\lambda_1$, the value of *SAE* could also be used to reflect particles size with larger particles showing a smaller *SAE* (Carrico et al., 1998).

The asymmetry parameter (*g*) is obtained from measured backscattering fraction according to the empirical function from Andrews et al. (2006).

$$g = -7.143889 \cdot \beta^3 + 7.4633439 \cdot \beta^2 - 3.9356 \cdot \beta + 0.9893, \tag{2}$$

where $\beta$ is the hemi-spherical backscatter fraction ($\sigma_{bsca}/\sigma_{sca}$) measured by the Aurora3000.

The absorbing coefficient ($\sigma_{abs}$) at different wavelengths (370, 470, 520, 590, 660, 880, and 950nm) was measured by an Aethalometer (AE33, Magee Scientific Inc, USA) (Hansen, 2005). The flowrate of AE33 was maintained at 4 L/min below 3000 m. The shadowing effect of the AE33 was corrected by the two spot measurements with different attenuation (Drinovec et al., 2017). The multiple scattering artifact of the AE33 was corrected by measuring the ambient aerosol in parallel with a photoacoustic spectrometer (PASS3, DMT Inc, USA), and the latter is independent of the filter artifacts. The PASS3 was calibrated using the $NO_2$ and BC standard (Arnott et al., 2005). Fig. S2 shows the two weeks' ambient measurements between AE33 and PASS3 at three overlapped wavelengths. Multiple-scattering correction factor of 2.88 was consistently found at three λ, which was applied to correct the AE33 measurement.

The absorbing Ångström exponent (*AAE*), which can weight the absorption at different wavelengths, is calculated using power fitting function at seven wavelengths.

$$\sigma_{abs}(\lambda) = \sigma_{abs,0}(\lambda/\lambda_0)^{AAE}, \tag{3}$$

We estimated $\sigma_{abs}$ of BrC assuming that BC is the only absorber at λ=950 nm, then the absorption of BC

at other wavelengths was extrapolated by assuming an *AAE* of 1 (Kirchstetter et al., 2004; Lack et al., 2013; Massabò et al., 2015), and the contribution of BrC at each wavelength was obtained by subtracting the BC absorption from the total absorption (Schnaiter et al., 2005; Liu et al., 2015a). It should be noted that previous studies pointed out the $AAE_{BC}$ may be less than 1, thus assuming $AAE_{\mathbf{BC}}$=1 may lead to an underestimation of BrC contribution (Gyawali et al., 2009; Lack and Cappa, 2010; Feng et al., 2013). We therefore consider the results reported here is the lower bound for the BrC contribution.

The single scattering albedo (*SSA*) is the ratio of the scattering coefficient over the extinction coefficient ($\sigma_{ext}$) at a given wavelength.

The parameters $\sigma_{sca}$, $\sigma_{abs}$, and $\sigma_{ext}$ are reported for standard temperature and pressure (STP, 1013.25 hPa, 273.15K) to allow for direct comparison at different altitudes among flights. Note that to compare with the AERONET results and for the radiative transfer calculations (as detailed in the following), these parameters in ambient conditions are used.

Column aerosol optical properties during the aircraft observation period were obtained from the Aerosol Robotic Network (AERONET) sun-photometer network (Che et al., 2009; Xia et al., 2008), where the site (AERONET BEIJING_PKU) is about 10 km away from the location of the vertical profiles. The measurement of $\sigma_{ext}$ was up to 2500m above which the aerosol concentration was low enough to be below the instrument lower detection limit. Given the very low concentration above 2500m, the value at 2500m was used to reconstruct the vertical profile up to 5000m. After that the $\sigma_{ext}$ from 2.5-5 km only accounted for 1-2 % of the integrated columnar extinction.

To evaluate the potential influence of the particle hygroscopicity on optical properties, the aerosol hygroscopic growth parameterization ($f$(RH)) is used to calculate the enhancement of $\sigma_{sca}$ under ambient RH. This function was previously measured by Zhao et al. (2019b) over Beijing region, expressed as:

$$f(RH) = a \cdot (1 - RH/100)^{-\gamma(RH/100)} \tag{4}$$

where $f$(RH) was obtained by a comparison between a dry and humidified nephelometers in parallel. a / γ was 0.930 / 0.329, 0.971 / 0.372, and 0.988 / 0.356 for clean, moderate, and heavy pollution period, respectively, according to the study.

The RH influence on $g$ was calculated according to Zhao et al. (2018), expressed as:

$$g(RH)/g(RH < 40\%) = a \cdot (1 - RH/100)^{-\gamma(RH/100)} \tag{5}$$

where a / γ was 0.9984 and 0.0849.

The $\sigma_{sca}$, $\sigma_{ext}$, *SSA,* and $g$ are all calculated for the hygroscopicity influence.

## 2.2 Radiative transfer calculation

The atmospheric irradiance and actinic flux using the pseudo-spherical version of the Discrete Ordinates Radiative Transfer Code (DISORT), as implemented in the libRadtran software package (Emde et al., 2016). The in-situ measured vertical profiles of AOD, single scattering albedo (*SSA*), and $g$ are used as inputs, and the other input parameters for the radiative transfer calculation are summarized in Table 2. The calculation is performed for clear-sky condition only, thus the flights experiencing low-level clouds are not included in the calculation. The direct, upward diffuse, and downward diffuse irradiance and actinic flux (AF, in $mWm^{-2}$) at $\lambda = 250\text{-}2550$ nm are calculated. The calculation of AF is performed with and without the aerosol input (AOD is set to zero) to evaluate the aerosol net impact. The heating rate is only calculated with considering the in-situ measured AOD. The spectral instantaneous absorbing power of BC ($A_{BC}$) or BrC ($A_{Brc}$) can be calculated by multiplying the absorption coefficient of BC (or BrC) and AF at specified $\lambda$, then integrating all $\lambda$ will obtain the total absorbing power (Gao et al., 2008; Emde et al., 2016), expressed as:

$$A_{BC\ or\ BrC} = \int_{250nm}^{2550nm} AF(\lambda) \cdot \sigma_{BC\ or\ BrC}(\lambda)\, d\lambda, \tag{6}$$

By assuming no radiative loss of solar energy and that the heat absorbed by aerosol is fully transferred to the surrounding air, the instantaneous heating rate of BC or BrC to the ambient air is hence calculated as:

$$H_{BC,BrC} = A_{BC,BrC}/(\rho \cdot C_p), \tag{7}$$

where $\rho$ and $C_P$ are the air mass density ($kg/m^3$) and heat capacity (1.007 J/g*K), respectively. The profiles of aerosol optical properties influenced by hygroscopic growth (as calculated above) are also input in the

197 calculation to work out its influence on heating rates.

## 3 Results and discussions

### 3.1 Overview and the pollution events

Three pollution events from Nov. 15[th] to 18[th] (Case 1), Dec .10[th] to 12[th] (Case2), and Dec. 16[th] to 19[th]
(Case 3) in 2016 were captured. Figure 1 shows the temporal evolution of surface $PM_{2.5}$, AOD (AAOD)
constrained by the in-situ aircraft measurements and from the AERONET, and the vertical profiles of $\sigma_{ext}$
and wind information during the Case 1 pollution event. The other two events are shown in Fig. S3 and
Fig. S4. Aircraft vertical profiles were performed on a daily basis as the flight time being indicated by the
vertical bars (Fig. 1). Each pollution event was classified as pollution initialization, development and peak
pollution period, corresponding to the pollution levels as clean period (CP, $PM_{2.5,\ surface}$ < 35 μg/cm$^3$),
transition period (TP, 35 μg/cm$^3$ < $PM_{2.5,\ surface}$ < 200 μg/cm$^3$) and heavy pollution (HP, $PM_{2.5,\ surface}$ > 200
209 μg/cm$^3$). Three flights (20161117 12:00, 20161117 15:00, 20161118 12:00) experienced boundary layer
cloud (Fig. 1c), as indicated by the intensive extinction on top of the PBL. There were 3, 4 and 4 profiles
in clear sky condition for the CP, TP and HP period respectively (as detailed in Table 1). As Fig. 1b shows,
wind sheers in both wind speed and direction appeared on top of the PBL, consistent with the vertical
distribution of $\sigma_{ext}$ observed by a lidar (Fig. 1c). During the CP, wind profiles (Fig. 1b) showed dominant
northwesterly wind with high wind speed throughout the column, dispersing the pollutant in a more
developed PBL (Fig. 1c). During the TP, the southerly air flow dominated and the $PM_{2.5}$ mass
concentration underwent a rapid increase from 30 to 100 μg m$^{-3}$ in several hours. During HP, the
windspeed was relatively low at all altitudes, maintaining the $PM_{2.5}$ mass concentration at a high level.

Figure 2 summarized the in-situ measured meteorological parameters at different stages of pollution
events. The height of PBL (PBLH) was determined by considering a variety of factors. Firstly, a stable
potential temperature ($\theta$) (Fig. 2d-f) with vertical gradient $d\theta/dz$ < 5 K/km in the PBL indicated an
sufficient convective mixing (Su et al., 2017), with a pronounced positive gradient above the PBL
indicating a stable layer (Petra Seibert, 2000). Secondly, there was usually a temperature inversion on top

of the PBL (Fig. 2a-c). During the CP, the weak temperature inversion (~0.15K/100m) on top of the PBL allowed pollutants to penetrate the PBL and disperse in a higher atmospheric column (Fig. 2b). This inversion was significantly enhanced for the TP and HP periods, to 0.9K/100m and 0.7K/100m respectively. The large increase of the inversion during flight 20161211 was caused by the regional transport from the south, when the lower-latitude warmer air mass was imposed onto the measurement level (Tian et al., 2019). Additionally, the PBLH decreased gradually as the pollution continued during the pollution event, in line with the enhanced aerosol concentration in the PBL. The moisture had similar features that a lower moisture content showed when lower pollution level and was vertically efficiently dispersed; whereas the stronger inversion trapped the moisture inside the PBL, leading to a positive vertical gradient with the maximum RH showing on top of the PBL. There were some regional transport influences under TP, resulting in enhanced RH when airmass was advected from the south (Fig. 2f).

## 3.2 Vertical profiles of $\sigma_{ext}$, $\sigma_{sca}$ and $\sigma_{abs}$

Figure 3 shows the vertical distributions of aerosol optical properties including extinction ($\sigma_{ext}$), scattering ($\sigma_{sca}$) and absorbing ($\sigma_{abs}$) coefficients. Different structures of vertical profiles were observed for CP, TP and HP periods. During the CP, aerosol concentration was low and showed a uniform mixing inside the PBL, with the $\sigma_{ext}$, $\sigma_{sca}$ and $\sigma_{abs}$ ranging from 220-270 Mm$^{-1}$, 180-240 Mm$^{-1}$, and 30-50 Mm$^{-1}$, respectively. The backward trajectories for the CP showed that the air masses were from the northwestern low emission region (Fig. S5). TP showed about a 4-fold increase of $\sigma_{ext}$ compared to the CP. During the TP, the $\sigma_{ext}$, $\sigma_{sca}$ and $\sigma_{abs}$ had large variation inside the PBL, ranging from 325-1435 Mm$^{-1}$, 300-1275 Mm$^{-1}$, and 45-160 Mm$^{-1}$, respectively, and the mean PBLH decreased to 200-500 m. During these pollution accumulation periods (before the pollution reached the peak level), two contrast vertical structures was observed. One showed a well-mixing in the PBL but declined concentration in the free troposphere (FT) (e.g. flight 20161115PM and 20161210) (Fig. 3a). The other one had the increased aerosol layer on top of the PBL, and showed positive vertical gradients for all optical properties at a certain level (e.g. flight 20161116 AM, 20161211 and 20161216) (Fig. 3b). The former was because of the mostly cleaner northwesterly air mass and higher wind speed influencing the layer above the PBL, while the latter

resulted from the southwesterly regional transport (Tian et al., 2019).

During the HP period, most flights consistently showed the exponentially-declined vertical profile patterns, and the PBLH was even lower than that during the TP (Fig. 2f). The stronger temperature inversion (Fig. 2c) and lower wind speed (Fig. 1b) inside the PBL led to high stability of the PBL and promoted the pollutant accumulation. The aerosol concentration was largely enhanced towards the surface and sharply declined above the PBL. Interestingly, the absorption showed a higher degree of negative vertical gradient than the scattering at $\lambda$=440nm, which may reflect the different sources and mixing ratios of absorbing and non-absorbing aerosols. The surface emission tends to contain more primary sources of absorbing particles such as BC and BrC, while enhanced secondary aerosol formation at upper altitudes may add additional aerosol extinction.

The vertical profiles of $\sigma_{sca}$ and $\sigma_{abs}$ during HP can be fitted as:

$$\sigma_{sca} = \sigma_{sca,0} \cdot \exp(-a * H) ; \quad a = 0.0012 \pm 0.0001, \tag{8}$$

$$\sigma_{abs} = \sigma_{abs,0} \cdot \exp(-b * H); \quad b = 0.0015 \pm 0.0001, \tag{9}$$

where $\sigma_0$ represents the surface value of $\sigma_{sca}$ and $\sigma_{abs}$, and $H$ is the altitude. The $a$ and $b$ are the parameters defining the changing rate with altitude. This parameterization could be used to represent the vertical structure of optical properties under heavy pollution condition.

The hygroscopic effect on aerosol vertical profiles was mainly controlled by the ambient RH (shown in blue lines in Fig. 3). For most of the flights, the hygroscopic effect could be neglected due to low RH (< 50%) (Fig. 2). For some of the flights (20161211), the $\sigma_{sca}$ and $\sigma_{ext}$ especially at top of the PBL could be enhanced by a factor of 1.3.

## 3.3 Vertical profiles of *SSA*, *SAE*, *AAE* and *g*

Figure 4 shows the vertical profiles of *SSA, SAE, AAE,* and *g* for all the flights during different stages of pollution events. Overall, the *SSA* showed two modes inside the PBL. During the CP, *SSA* for most flights was populated at 0.85, and had less variation throughout the column in the PBL. Flight 20161115AM

showed a strong elevation of *SSA* (0.94) at 2200 m (Fig. 4a), which may be influenced by a dust layer (to be further discussed below). *SSA* showed a positive vertical gradient for the TP and HP inside the PBL, i.e. from the surface to the PBLH, the mean *SSA* increased from 0.85 to 0.91 and from 0.87 to 0.92 for the TP and HP period, respectively. This indicates the reduced fraction of absorbing particles, in turn suggesting an enhancement of secondary production for non-absorbing particles. There were a few profiles featuring with a large enhancement of *SSA* (>0.95, for flight 20161211) at high altitude (Fig. 4b), and backward trajectory analysis (Fig. S5) showed that these resulted from the regional transport when more aged pollutants were advected to the high altitude. The *SSA* in the FT was mostly higher than that in the PBL and maintained at 0.9-0.95 for the TP and HP, suggesting a lower absorbing particle fraction at higher altitudes. Comparing among different stages during pollution events, it could be concluded that at the initialization stage of the pollution when the total PM was relatively low, a lower *SSA* exhibited, while the increase of pollution level added more secondary substances, hence increasing *SSA*. This trend was consistent with the previous ground studies in Beijing (He et al., 2009; Jing et al., 2011).

The *SAE* reflects the particle size with larger size having a smaller *SAE*. A decreasing *SAE* was shown for increasing pollution levels inside the PBL (Fig. 4), i.e., from *the* CP to HP, the *SAE* in the PBL showed an average value of 1.74, 1.45, and 1.21, respectively. For most of the profiles, *SAE* showed enhancement at higher altitudes. This means smaller particle sizes were present at higher altitudes, which may result from a higher scavenging efficiency for larger particles whereas smaller particles remained un-scavenged at the upper height (Liu et al., 2009). These was an exception for flight 20161211, when the regional advection transported larger and aged particles to the higher altitude. The particle size also corresponded with the asymmetry parameter (*g*, Fig. 4j-i), with larger particle representing more fraction of forward scattering (larger *g*). Note that there was only one flight (flight 20161211) under RH > 80 %, where the particle hygroscopicity had appreciable influences on *SSA* (increased by 0.05), *SAE* (decreased by 0.2) and *g* (increased by 0.1).

*AAE* reflects the degree of the absorption towards shorter wavelength, such as the presence of BrC will enhance the absorption in the UV. A lower *AAE* of 1.2 ± 0.2 was shown for the CP (Fig. 4g), but it increased to 1.56 ± 0.3 for TP in the PBL (Fig. 4h), and an additional higher mode of *AAE* showed at 1.8-2.0 for the HP period (Fig. 4i). There was a weak variation of *AAE* for CP throughout the column, but

became largely spread for TP, i.e., with either positive or negative vertical gradient at different altitudes. Notably, the *AAE* showed consistent positive vertical gradient for most of the HP profiles (Fig. 4i). This implied the enhancement of BrC contribution at higher altitude for the polluted troposphere. Flight 20161115AM showed a notably increased *AAE* up to 2 at altitude 2 km (Fig. 4g), which may reflect the influence of dust (Cazorla et al., 2013). The ground *AAE* had strong seasonal variation with winter normally showing a higher *AAE* due to higher emissions of solid fuel burning (Sun et al., 2017; Wang et al., 2018b). However, there is still lack of results on the vertical characteristics of *AAE* due to limited measurements, and the results here highlight the enhancement of BrC at higher level, mainly for polluted environment.

### 3.4 Comparison of column integrated and in-situ constrained AOD/AAOD

To compare the AOD and AAOD between the AERONET and that constrained by the in-situ measurements, the AERONET data was chosen to match with the aircraft profiles in time (±3h) and location (within 10 km) (the PEK site). The comparison was performed at overlapped wavelengths (440nm, 675nm, and 870nm) between AERONET and aircraft instruments. As Fig. 5a-c shows, high correlation ($R^2 > 0.95$) was found between the columnar and in-situ measurements. In particular, the correlation was close to unity for dry conditions (RH < 40%), while the AERONET data was about 10-20% higher than the in-situ measurement for RH > 60%. Improved agreement was achieved by 8-15% if considering aerosol hygroscopic growth (open circle in Fig. 5a-c), despite that the in-situ constrained AOD was still 2-5% lower than the AERONET after the hygroscopic correction.

Figure 5g-i shows at three wavelengths the AAOD had lower correlations between both methods compared to AOD, with $R^2 = 0.75$, 0.58, and 0.49 at 440 nm, 675 nm, and 870nm, respectively. The columnar AAOD was overall about 10-25% higher than the in-situ measurement, and this AERONET AAOD overestimation was higher under higher AOD conditions. This is consistent with previous findings conducted over the USA that the AAOD retrieved from AERONET was biased higher when compared to the in-situ measurement (Andrews et al., 2017).

Note that there was better agreements during the CP, when lower pollution level and lower RH (shown in blue dots). This suggests a lower moisture and less AOD interference may improve the agreement on AAOD. Previous studies pointed out that the retrieval from the AERONET was sensitive to the variation of aerosol vertical distribution (Torres et al., 2014). We thus speculate that the better agreement for CP was due to the vertically homogeneous distribution of aerosol optical properties, and the larger bias for the TP and HP periods might be caused by the significant variations of the vertical profiles. Other factors like the aerosol hygroscopic growth under higher RH may introduce factors in enhancing the absorption, e.g. more lensing effects on BC absorption via thicker and moisture coatings (Wu et al., 2017). Though this study is not able to rule out the exact influencing factor in causing this discrepancy, an overestimation of 25% in the AERONET AAOD under polluted conditions is shown for the dataset here.

**3.5 Heating impacts of BC and BrC**

Figure 6 shows vertical profiles of irradiances from the radiative transfer calculation using in-situ measurements as model inputs (Table 1). The results show that the presence of aerosols reduced the direct irradiance reaching the surface (Fig. 6a-c) but increased the upward diffuse (Fig. 6d-f) and downward diffuse irradiances, especially above the PBL (Fig. 6g-i). The direct irradiance on the surface ranged from $1 \times 10^9$ to $3.5 \times 10^9$ mW m$^{-2}$, with an average of $2.2 \times 10^9$ mW m$^{-2}$ during the CP (Fig. 6a), which was about two-fold and three-fold larger than that during the TP (Fig. 6b) and HP period (Fig. 6c), respectively. The combined direct, diffuse upward and downward irradiances which forms the actinic flux (AF), showed an enhancement above the PBL and a reduction within the PBL (Fig. 7a-c), but to what extent the enhancement or reduction occurred depended on the aerosol vertical profiles. The vertical gradient of AF was slightly modified by the aerosol loadings during the CP, whereas for the TP and HP, aerosol effects caused AF about two times smaller within the PBL and 20 % larger above the PBL, leading to an increased vertical gradient of AF. The AF received at lower altitude was reduced by up to 10 % by incorporating the aerosol hygroscopicity influence (Fig. 7) due to the enhanced AOD, and AF was further redistributed to give a larger vertical gradient (Fig. 7a-c).

The vertical profiles of absorbing power and heating rate of BC are shown in Fig. 7d-f. Vertically homogeneous BC heating rate of 0.05 K/h was found inside the PBL during the CP (Fig. 7d). During the

regional transport cases (flight20161211 and flight20161216) for the TP, positive vertical gradients (increase with increasing altitude) of BC heating rates were observed, and as high as 0.1 K/h heating rate could occur at top of the PBL (Fig. 7e). During the HP period, negative heating rate (decrease with increasing altitude) of BC was found except for one flight on 20161212 in Case 2, and the BC heating rate at the surface could reach as high as 0.15 K/h (Fig. 7f). The reason causing the negative vertical gradient of BC heating rate was the higher degree of negative gradient of $\sigma_{abs}$ (Fig. 3i) than the positive gradient of AF (Fig. 7). The results here show that the atmospheric heating by aerosol was mainly inside the PBL and for polluted period the BC-induced heating was 0.05-0.17 K/h, generally consistent with previous studies over the polluted Asia region, with 0.02-0.17 K/h (Ramana et al., 2007; Ramana et al., 2010; Kedia et al., 2010).

The contribution of BrC to absorbing power and heating rate was computed as the integrated portion of absorption over the visible wavelength (370–950nm in this study) by subtracting the BC absorption from the spectrum. Figure 7g-i shows the vertical profiles of BrC heating rate. Continuously increase of BrC heating rate in the PBL was observed from CP to HP, with mean heating rate of 0.02 K/h, 0.03 K/h, and 0.05 K/h during the CP, TP and HP respectively. Though the BC was the main contributor to the heating in the PBL, the heating of BrC was more evenly distributed and could be comparable with the BC heating rate at high altitude especially during the HP period (Fig. 7i). The contribution of BrC to the total absorption was reported to be 10-27 % over the polluted region of Europe (Ferrero et al., 2018) and south Asia (Chung et al., 2012; Shamjad et al., 2015), in general consistent with the results during polluted periods here.

Corresponding with the aerosol hygroscopicity influence on the actinic flux, the heating rate showed lowered intensity but enhanced vertical gradient for the flights with high ambient RH (Fig. 7b). The vertical gradient of the overall heating rate from absorbing components, i.e. increase or decrease heating rate with altitude, will importantly determine the influence on atmospheric stability. If the heating occurred near surface (Case 3), the lower layer will be heated leading to an enhanced convective mixing (Sühring et al., 2014; Petaja et al., 2016); whereas if the heating was above the PBL (Case 2), an increase of temperature inversion will occur hence inhibiting the PBL development and trapping the pollutants in the PBL (Chakrabarty et al., 2012; Tripathi et al., 2007). This study showed positive vertical gradients for

30 % of the flights especially during the regional transport when pollutants were advected from outside of Beijing and showed elevation of absorption at higher altitude (Fig. 8). The rest of the flights showed highly accumulated aerosol concentration near surface, and BC potentially promoted the dispersion in the PBL and decreased its stability. This was also found by a previous study (Ferrero et al., 2014).

**3.6 The importance of BrC heating effects**

Figure 8 shows the measured absorption coefficient of BrC and BC inside and above the PBL at different $\lambda$ for CP, TP and HP period, respectively. The results suggested that both $\sigma_{abs}$ of BC and BrC increased with the pollution level, e.g. the $\sigma_{abs}$ at $\lambda$= 440nm was 42.8 Mm$^{-1}$ and 7.2 Mm$^{-1}$ on average in the PBL and above the PBL respectively during the HP, and was 4.7 Mm$^{-1}$ and 1.3 Mm$^{-1}$ for the CP. The contribution of BrC to total $\sigma_{abs}$ was found to increase from the CP to the HP period (Fig. 8c, f). This is in line with previous studies in urban Beijing that more BrC contribution to total absorption was found under higher pollution level (Ran et al., 2016b; Xie et al., 2019), suggesting the important role of BrC on absorption under polluted condition.

The contribution of BrC to total heating rates showed notably different vertical structures. During the CP, all profiles showed consistently low BrC contribution throughout the column, with about 7 % at the surface and 9 % in the FT (Fig. 8g). This means the low primary emission or the emission after being diluted by clean air mass did not contained a large fraction of absorbing organics. During the TP, BrC contribution inside the PBL increased to 22 % and showed considerable variations at higher altitude (Fig. 8h). During the HP period, the surface contribution was comparable with that in TP, but showed remarkably enhanced BrC heating contribution at higher altitudes, with a vertically increasing rate of 1.5 %/m in the PBL and reached as high as 45 % in the LFT. The higher heating contribution of BrC at higher altitudes means the BrC absorption played an important role in heating at upper altitudes, which may enhance the temperature inversion at that level hereby inhibiting the convective mixing under the heated layer.

By comparing the BrC heating contribution at the surface, there was an increase from CP to TP, however, not from TP to HP. This suggests the primary emission will increase the BrC fraction from CP to TP, but

for even more polluted environment from TP to HP, the primary emission may provide limited further increased faction of BrC. The primary BrC could result from a range of combustion sources, and the polluted region at the south of Beijing contained a higher fraction of residential coal burning sources (Sun et al., 2017; Xie et al., 2019) which may influence the Beijing region under polluted period in wintertime. The relatively consistent BrC contribution at ~20 % from TP to HP suggested the relatively uniform BrC profiles for the primary sources. During the TP, the BrC contribution above the PBL had rarely been above 30 % (Fig. 8h), however during the HP, there was a further enhancement of BrC contribution up to 45 % above the PBL (Fig. 8i). Note that there was no direct injection of biomass burning plume to the high altitude during the study period, the higher portion of BrC absorption above the PBL during HP may be formed through secondary production in addition to the primary source contribution. As Fig. 7a-c shows, there was more intensive actinic flux received at higher altitude and this may promote the photochemical reactions of gas-phase species, allowing more secondary formation of aerosol which may contain a fraction of BrC (Feng et al., 2013; Nakayama et al., 2013). Previous studies also found enhanced BrC formation with light source under certain RH (Nguyen et al., 2012; Updyke et al., 2012; Laskin et al., 2015; Zhao et al., 2015). The positive gradient of BrC heating contribution was more likely resulted from the enhanced RH from the surface to the top of PBL (Fig. 2i), because increased moisture will promote the aqueous reaction and gas-to-aerosol partition which may also form part of the BrC observed here (Ervens et al., 2011; Nakayama et al., 2013). The secondary formation of BrC also requires the inorganic or VOC precursors being transported to the high altitudes, therefore the enhancement of BrC mostly occurred under higher pollution level when sufficient gas precursors were transported to higher altitudes. The BrC may be also subject to bleaching process and lose the absorbance (Sareen et al., 2013; Lee et al., 2014; Wong et al., 2019), because the profiles in this study were conducted over an urban megacity where the sampled pollutants were fairly young and may have not experienced sufficient ageing for BrC to be degraded.

## 4 Conclusions

This study provides detailed characterization of vertical profiles of aerosol optical properties over the Beijing region by continuous aircraft in-situ measurements at different stages during the pollution events.

The results combining direct measurements of scattering and absorption at multiple wavelengths, give a full picture of how the optical properties had evolved at different layers during typical pollution events. During the clean period for pollution initialization (CP), the aerosols showed relatively uniform characteristics throughout the planetary boundary layer (PBL) and lower free troposphere (FT), such as a lowing scattering or absorption coefficient, larger $SAE$ (due to smaller particle size) and lower fraction of brown carbon (BrC), as reflected by a smaller $AAE$. The transition period (TP) when pollution was developing had large variations of all optical properties, and enhanced aerosol loadings at higher altitude were encountered when being influenced by regional advection. The fully developed heavy pollution period (HP) featured with the shallow PBL accumulated over 80 % of the scattering and absorption within the PBL, and deceased $SAE$ due to the enlarged particles size. Notably the absorption towards shorter wavelength became larger under more polluted environment, especially at the higher altitude.

The AOD and AAOD measured by the passive remote sensing was for the first time compared with in-situ measurements over this polluted region. AOD showed high correlation between the AERONET and in-situ measurement within 10 %, and the most discrepancy between both could be possibly resolved by considering the hygroscopic growth of aerosols under high RH condition. The AAOD however showed 10-25 % higher for the remote sensing especially at shorter wavelength, consistent with other studies (Müller et al., 2012; Andrews et al., 2017). The possibilities of causing this could be the non-homogeneously vertically structures of optical properties, mixing state of light-absorbing aerosol, and also the particle hygroscopic growth, which are unable to be elucidated only using the results here.

BC was the main heating species, inducing 0.05 K/h, 0.1 K/h and 0.15 K/h heating rate at local times from 12:00 to 15:00 in the PBL during pollution initialization, transition and full development, respectively. The heating rate showed positive vertical gradient during the regional transport period when pollution was advected to a high altitude from the polluted southern region of Beijing (Tian et al., 2019). The contribution of BrC to the heating rate was found to increase by 20 % throughout the column from the CP to the HP period, in particular the increased BrC contribution was pronounced at the layer above the PBL during the HP, which was proposed to result from intensive photochemical reactions above the PBL. The BrC present at this layer will have the potential to contribute to the heating, hence enhancing the temperature inversion on top of the PBL hereby its the capping effect to the pollutants. Particles at

higher altitude may be transported to wider regions spatially in both vertical and horizontal directions, which may lead the BrC present at this layer to have wilder and longer radiative impacts. Different mechanisms of BrC formation at different levels such as above the PBL (where more solar flux received) or within the PBL (where more moisture was constrained) warrants future study.

**Data availability.** All data in this paper are available from the authors upon request (tianping@bj.cma.gov.cn).

**Competing interests.** The authors declare no conflicts of interest.

**Author contribution.** D. D., and M. H., led and designed the study; P. T., and D. L., designed the study, set up the experiment, analyzed the data, and wrote the paper. P. T., D. Z., and Q. L., conducted the aircraft observation. C. Y., performed the radiative transfer model calculation. P. T., D. L., Z. D., L. R., and Y. W., contributed to the aircraft data analysis. S. D., and K. H., contribute to the surface data analysis. G. Z., and C. Z., conducted the aerosol absorption comparison experiment.

**Acknowledgment.** This research was supported by the National key Research and Development Program of China (2016YFA0602001), the National Natural Science Foundation of China (41875044, 41875167, 41675038, 41975177), and the Beijing Natural Science Foundation (8192021). Part of this work is supported by the National Center of Meteorology, Abu Dhabi, UAE under the UAE Research Program for Rain Enhancement Science.

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

Table 1. Flight summary in this study.

| Flight number | Time range Local time | Case | Pollution period | Mixing layer height |
|---|---|---|---|---|
| RF1 | 20161115 12:00 | Case_1 | CP | 1450 m |
| RF2 | 20161115 14:00 | Case_1 | CP | 1450 m |
| RF3 | 20161116 12:00 | Case_1 | TP | 850 m |
| RF4 | 20161116 14:00 | Case_1 | TP | 750 m |
| RF5 | 20161117 12:00 | Case_1 | TP(Cloud) | 1250 m |
| RF6 | 20161117 14:00 | Case_1 | TP(Cloud) | 1150 m |
| RF7 | 20161118 12:00 | Case_1 | HP(Cloud) | 1050 m |
| RF8 | 20161210 14:00 | Case_2 | CP | 950 m |
| RF9 | 20161211 14:00 | Case_2 | MP | 950 m |
| RF10 | 20161212 14:00 | Case_2 | HP | 450 m |
| RF11 | 20161216 14:00 | Case_3 | TP | 350 m |
| RF12 | 20161217 14:00 | Case_3 | HP | 350 m |
| RF13 | 20161218 14:00 | Case_3 | HP | 350 m |
| RF14 | 20161219 14:00 | Case_3 | HP | 250 m |

[1] CP, TP, and HP represents the clean, transition and high pollution period during a pollution event.

Table 2. Summary of input parameters for the radiative transfer calculation using Discrete Ordinates
Radiative Transfer Code (DISORT)

| Parameter | Input value |
| --- | --- |
| Radiative transfer solver | DISORT, 12-streams, delta-m method |
| Gas absorption parameterization | LOWTRAN/SBDART parameterization |
| Wavelength range | 250-2550nm |
| Atmosphere | Standard Mid-latitude atmosphere |
| Aerosol | The 25 layers from the surface to 5000 m was chosen inside the DISORT, and every 200 m average aerosol optical properties at 550 nm was used; AOD values are derived from in-situ Aurora3000 and AE33 measurements, also applying an exponential $\lambda$-dependent function SSA values are from in-situ $\sigma_{sca}$ and $\sigma_{abs}$ measurement SAE values are from in-situ Aurora3000 measurement AAE values are from in-situ AE33 measurement Asymmetry factor (g) is derived from the Aurora3000 measurement and uses Henyey-Greenstein phase function |
| Location | 39.54ºN, 116.23ºE |
| Time | Flight time |
| Solar zenith angle | Effective solar zenith angle Using local time and aircraft location |
| Surface albedo | IGBP surface type 13 (Urban) |

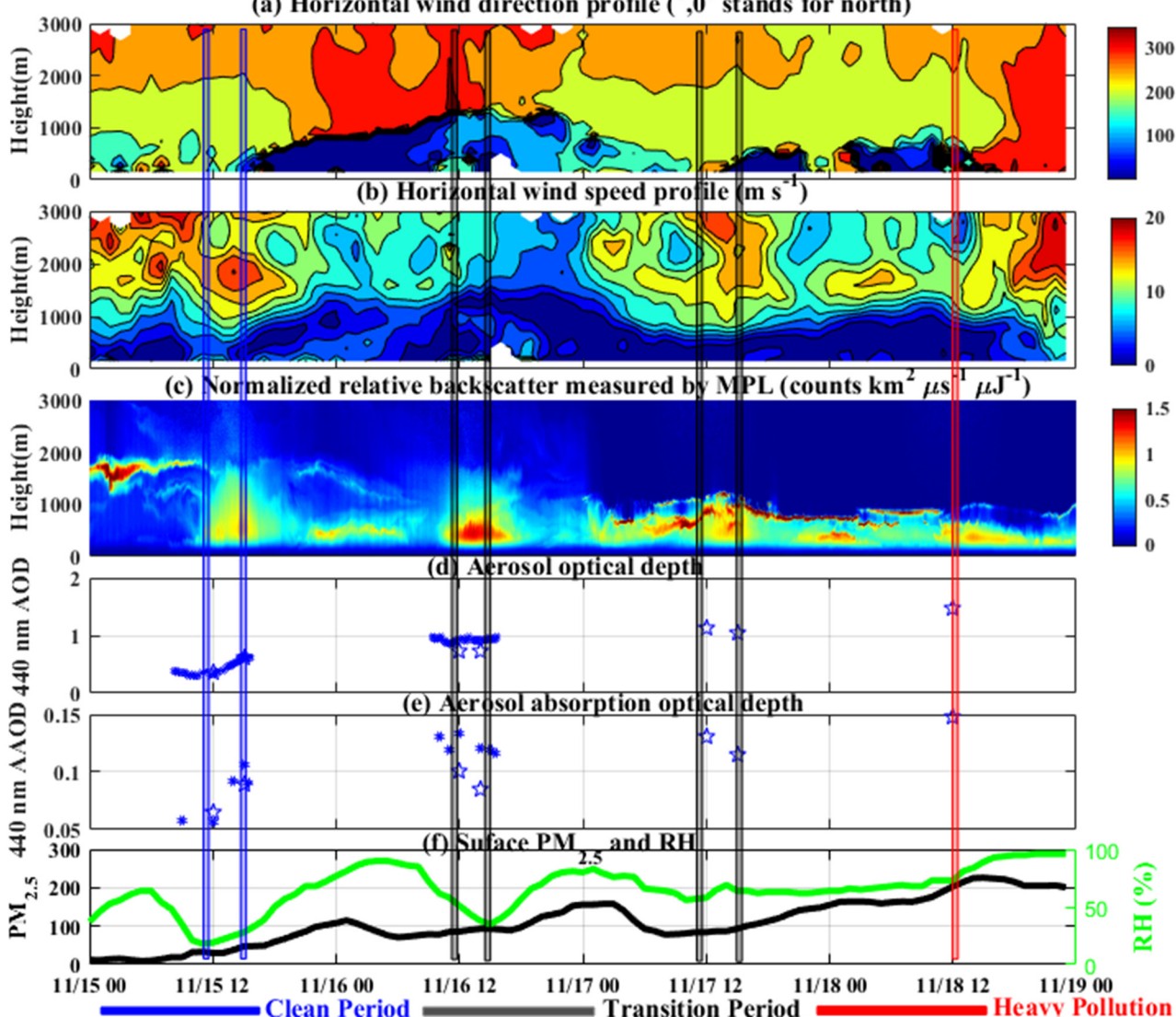

Fig. 1. Temporal variations from Nov. 15th to 18th of vertical profiles of wind direction (a), and wind

speed (b) measured by wind profile radar,; (c) particle extinction measured by MPL lidar; ) aerosol

optical depth (d) and aerosol absorption optical depth (e) from AERONET (asterisk) and derived from

aircraft in-situ measurements (open star)(f) surface PM2.5 and RH. ). The vertical bars denote the periods

of flight profiles, with blue, black and red representing the clean period, transition period and heavy

pollution during a pollution event, respectively. The other two pollution events are shown in the

← **Clean Period** → ← **Transition Period** → ← **Heavy Pollution** →

Fig. 2. Vertical profiles of temperature (a, b, c), relative humidity (d, e, f) and potential temperature (g,

813 h, i) for Clean Period, Transition Period and Heavy Pollution period, respectively. The black and red

dots represent for inside the PBL and above the PBL.

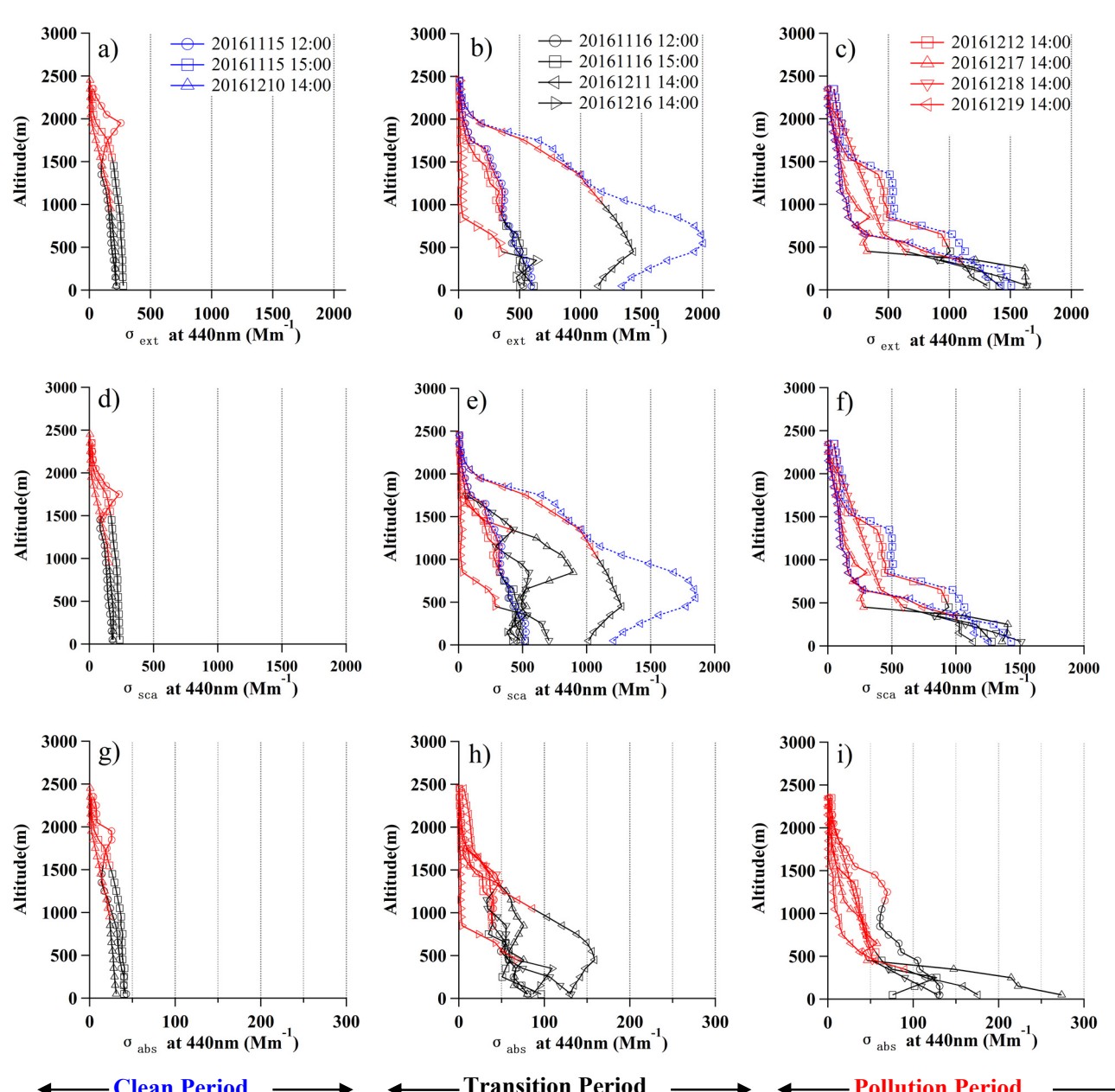

Fig. 3. Vertical profiles of aerosol extinction, scattering and absorption coefficient at 440 nm for CP (blue), TP (black) and HP period (red), respectively. The black and red lines represent for inside and above the ML, respectively. The hygroscopic-corrected profiles are shown as blue lines.

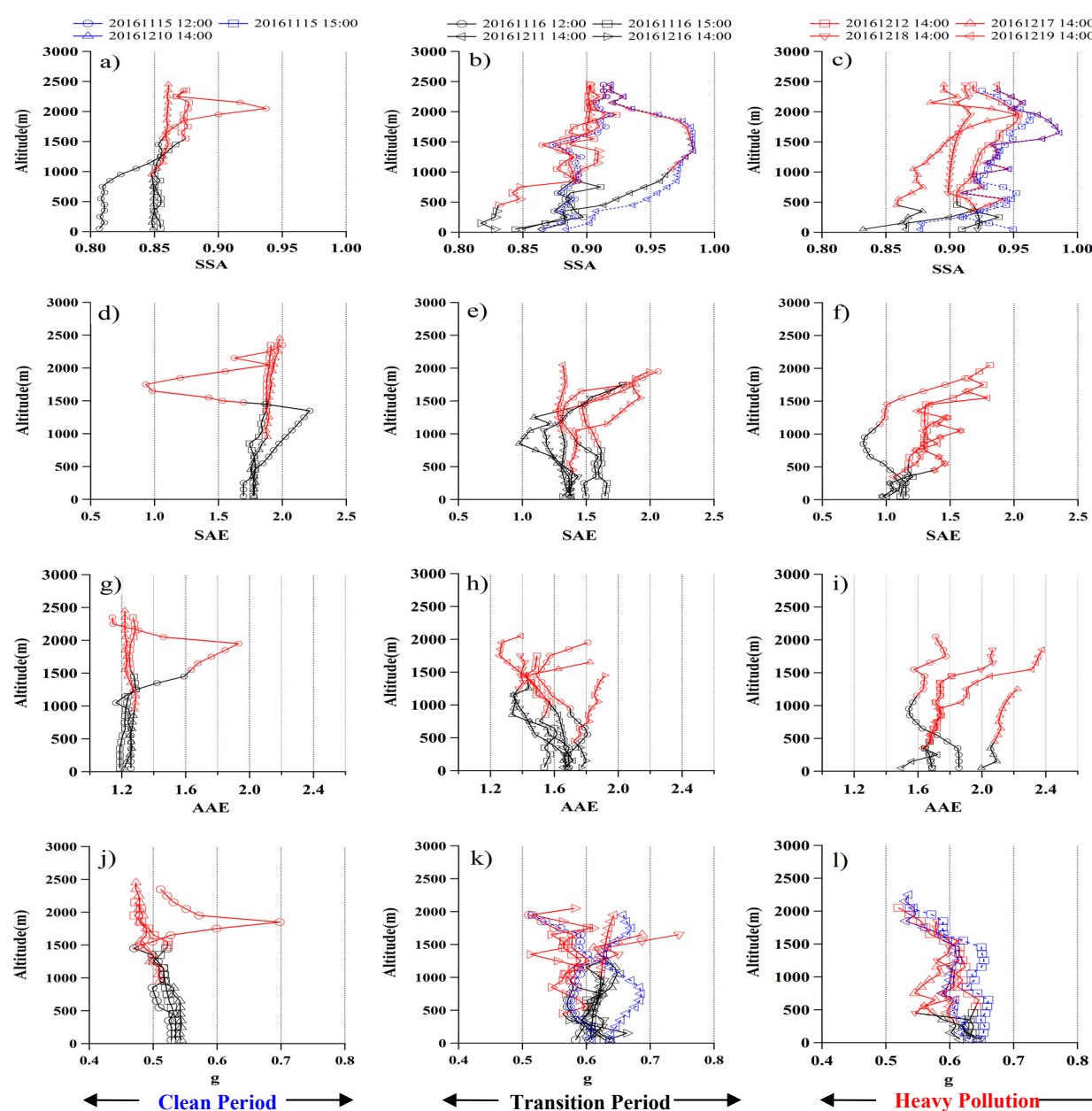

Fig. 4. Vertical profiles of aerosol single scattering albedo at 440 nm (SSA, a - c), scattering Angström exponent (SAE, d - f), absorption Angström exponent (AAE, g - i), and asymmetry parameter (g, j - l) for CP (left panel), TP (middle panel) and HP period (right panel), respectively. The hygroscopicity

corrected profiles was shown in blue lines.

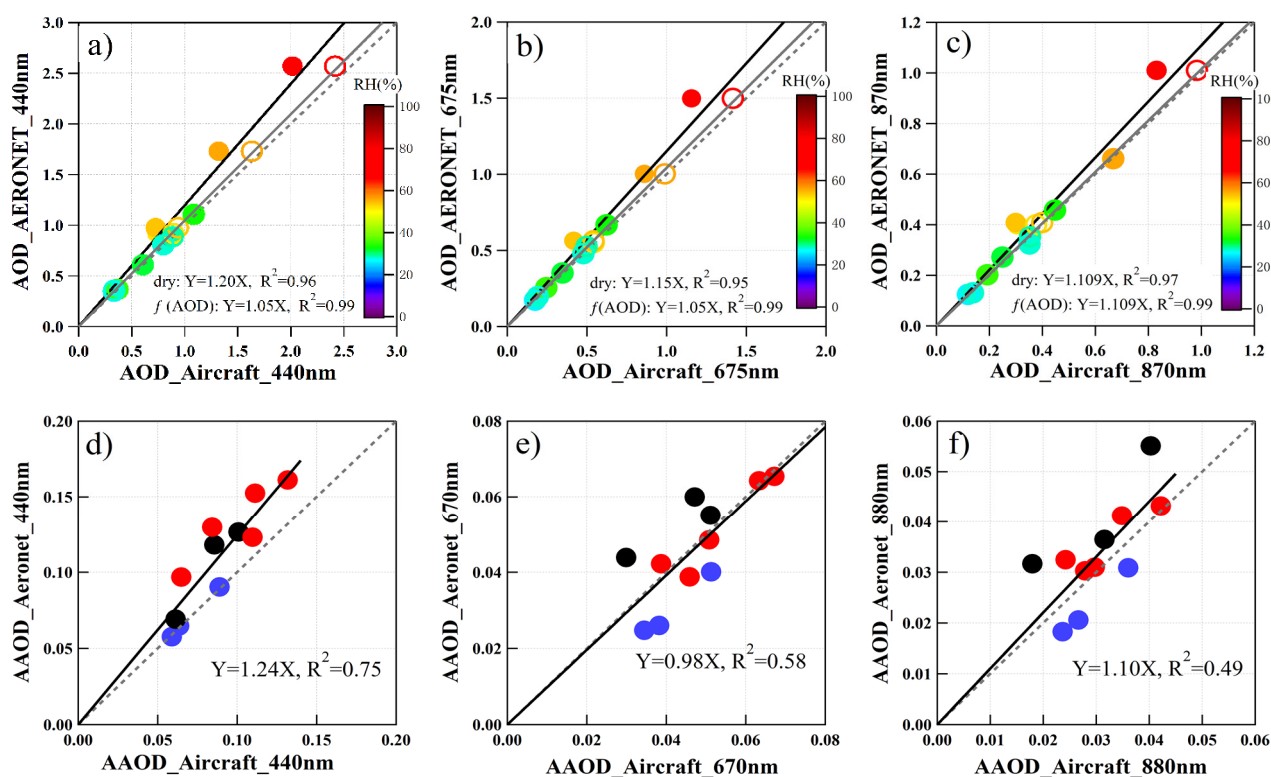

Fig. 5. Comparison between AERONET and aircraft in-situ constrained AOD and AAOD: a) - c) The comparison of AOD at 440nm, 675nm, and 870nm colored by RH; the solid and open markers denote the dry and hygroscopicity-corrected conditions; d) – f) Comparison of AAOD at 440nm, 670nm, and 880nm. The blue, black, and red circles represent the CP, TP, and HP periods, respectively.

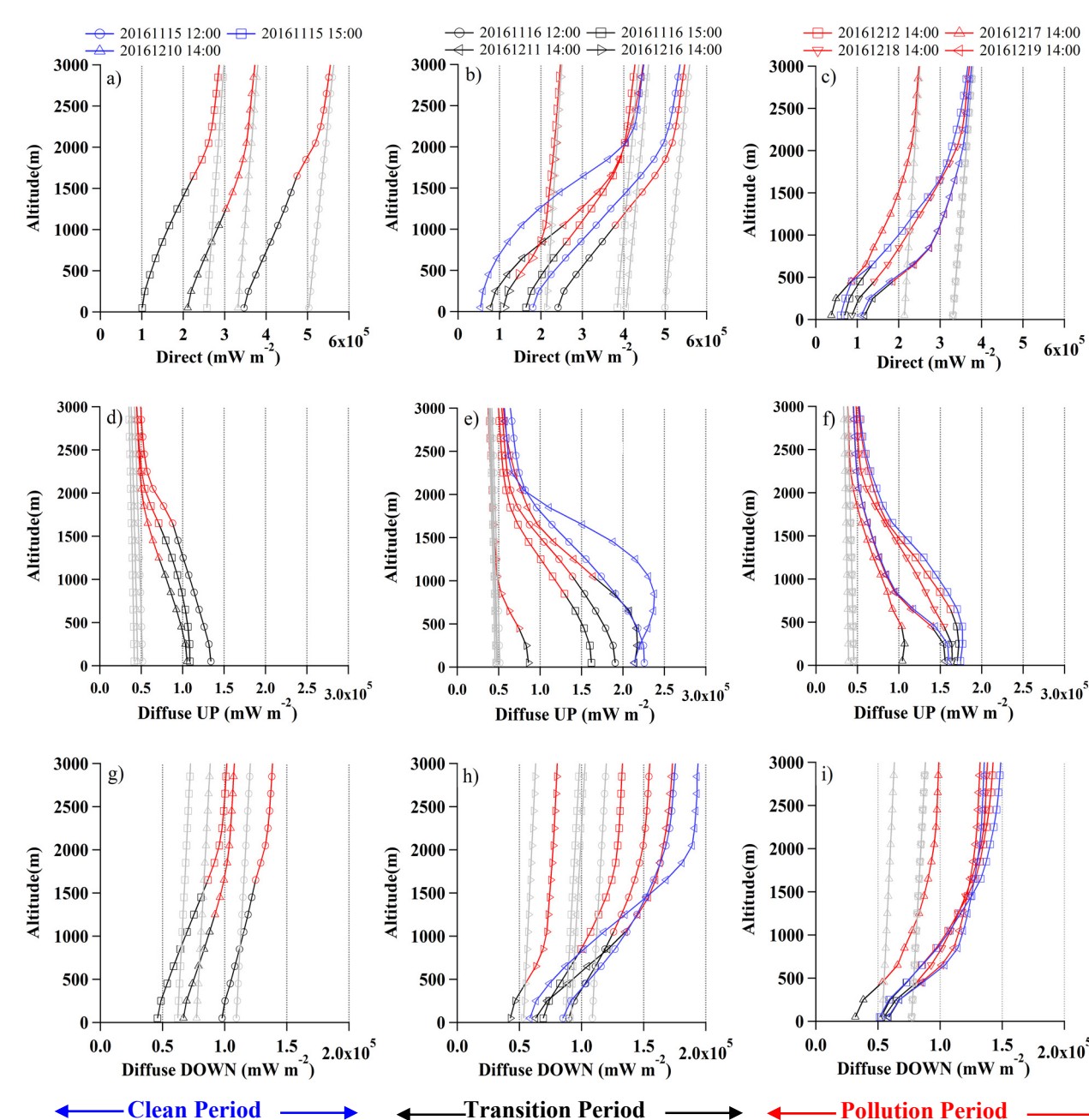

Fig. 6. Radiative transfer results calculated by DISTORT. a)-c), b)-e), and g)-i) show the direct, diffuse

upward and diffuse downward irradiance respectively. The left, middle and right panel represent for CP,

TP and HP period respectively, with black and red lines denoting above and within the PBL. The

colored and grey lines denote the profiles for with and without aerosol influence, respectively, and the

blue lines are for corrected hygroscopic effect.

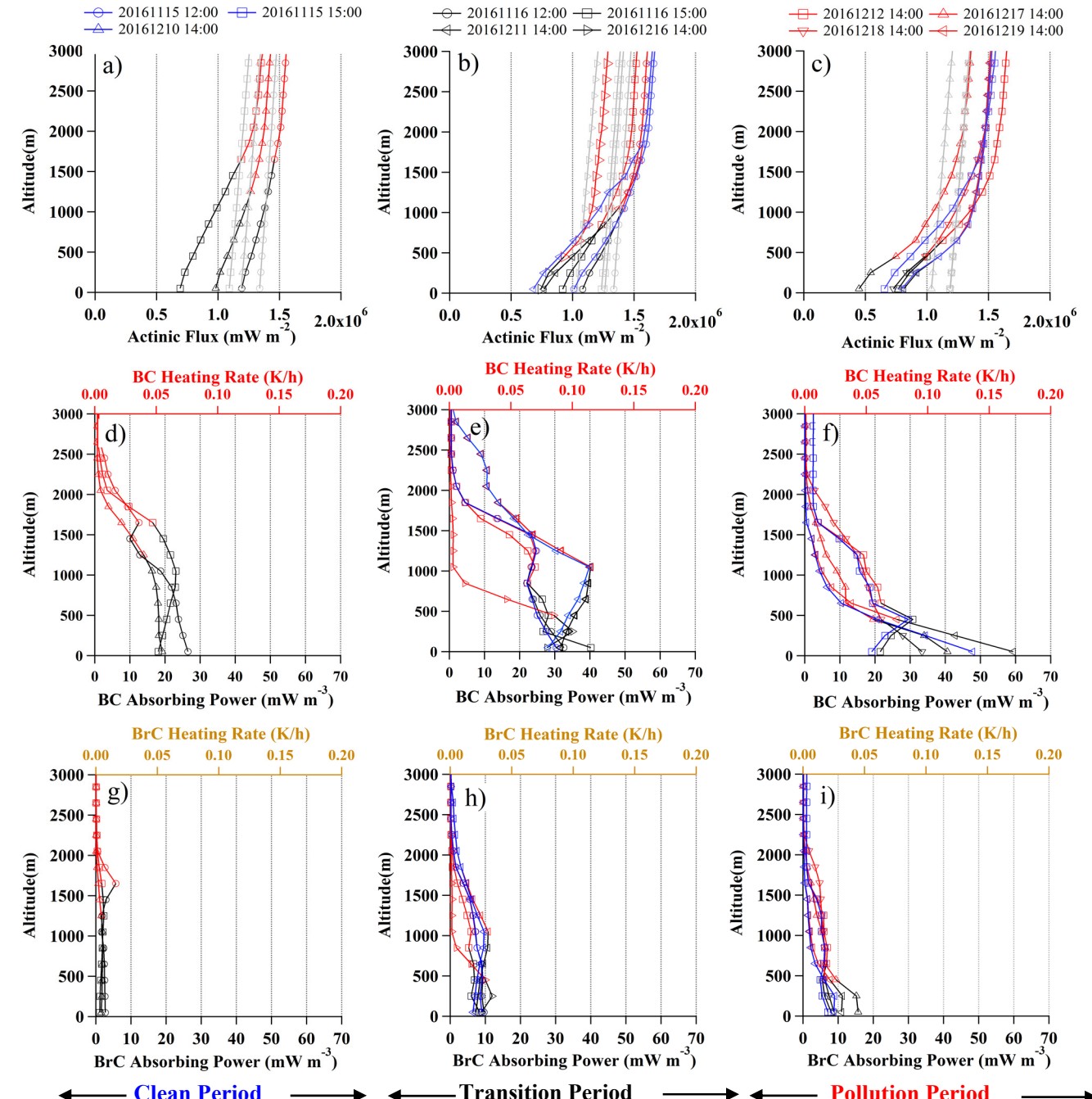

Fig. 7. Actinic flux (a-c), BC absorbing power (d-f) and BrC absorbing power (g-i). The left, middle and

993 right panel is for LP, TP and HP respectively, with the black and red line denoting within and above the

994 PBL. The gray lines in a) to c) show the aerosol free results and the blue line denote the corrected

hygroscopic effect. The upper x-axis from d) to i) shows the heating rate.

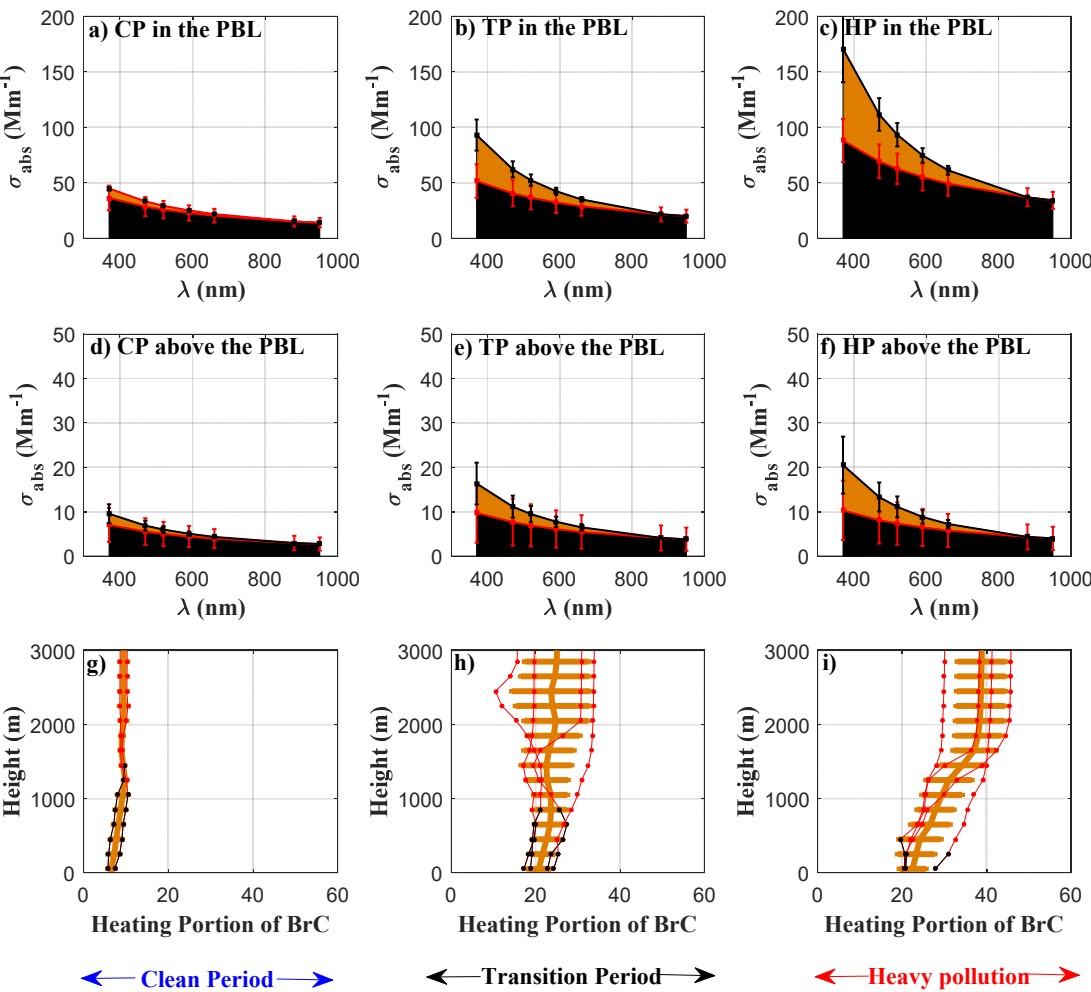

Fig. 8. Spectral absorption coefficient of BC and BrC inside and above the PBL for CP (a, d), TP (b, e)

and HP period (c, f), respectively, shown in black and brown carbon color respectively. The vertical

profiles of heating portion of BrC for CP, TP and HP period are shown in g) – i).
