# Peer review of "In-situ vertical characteristics of optical properties and heating rates 1"

_Atmospheric Chemistry and Physics, 2019_

## Referee Comment (RC1) · Anonymous Referee #1 · 16 Oct 2019

This manuscript presents a valuable observational dataset of in-situ aircraft measurement of BrC and BC optical profiles in Beijing. The corresponding influences on heating rate and radiative forcing are analyzed and extensively compared with AERONET dataset. Although the pollution and meteorology interaction over North China Plain (NCP) were widely investigated through surface observations and modelling, limited studies have considered the evolution of pollutants in vertical profile. This work could fill this gap well. The method and uncertainties are well described and discussed. The manuscript is well-written, but some parts of it are not clear enough. I would recommend for publication after the authors address the following specific comments:

Specific Comments:

1) line 22: "replying" or "relying" ?

[Figure]

2) line 68-69: "and regional transport will introduce enhanced aerosol loading to high level". Please introduce the corresponding vertical transport processes in more details base on previous studies over NCP, such as the influence of mountain valley breeze led by the special topography over Beijing.

3) line 73. Suggest delete "successive", which is too subjective.

4) line 77. Suggest delete "for the first time". There are lots of previous studies regarding aerosol optical property observations over NCP, although may not elaborate the detailed BrC properties as this work does. The "first time" description is not appropriate here.

5) line 86. At which temperature level are maintained?

6) line 93. Please give the criteria of screening out the "in-cloud data".

7) line. 116. "which is independent of the filter artifacts". I do not understand here.

8) Eq.3. Please modify it as the same format of the Eq.1.

9) line 152. Please specify the values (with units) of "air mass density" and "Cp" here.

10) line 269. Where was the study of Andrews et al. (2017) conducted?

11) line 384. They are the heating rate at which level? And are they the rate at noon time? Please clarify it.

12) line 386. "when regional transport". I do not understand here.

13) line 386. "contribution of BrC" to what? You mean contribution to the aerosol mass or heat rate or light absorption, or what?

14) Figure 1. The title of Fig. 1c is difficult to understand. And please add labels for the colorbar.

15) Figure 2 and following profile figures. Here, use blue, black and red to indicate the clean, transition and polluted period, respectively. However, it is ambiguous that in

one profile belongs to two different period (black for lower part, but red for upper part). In my understanding or the common understanding the "period" is separated by time windows.

16) Figure 4. The quality of this figure is poor and unreadable.

---

## Referee Comment (RC2) · Anonymous Referee #2 · 17 Oct 2019

This manuscript presents aerosol optical properties and heating rate along vertical profiles. Due to the several feedbacks triggered by the vertical forcing behaviour I encourage the publication of this work after a serious revision as some parts are not clear enough and other require an improved description. Finally, a comparison with other data available in literature is required. A final improvement of the english is mandatory due to the several typos present along the paper.

Major comments are reported here below:

1- Introduction and line 75-78: It is obviously clear that the focus of the paper is to describe the situation over Beijing, however a world-wide contextualization of existing heating rate data (along vertical profiles as well as at ground) is mandatory concerning the importance of this topic.
2- Section 2, Line 88 and section 3.4 lines 263-265: "silicate direr (change to dryer) instruments was utilized (better used) ahead all instruments to maintain the sampling RH lower than 40%" and "Improvedagreement between both may beachieved by considering the particles hygroscopic growth, which requires composition measurement to constrain this factor but this was not available inthis study". This is a serious lack in the work due to the aim to perform radiative transfer calculations. The choice to measure dry aerosol optical properties (especially for scattering) seriously affect the SSA, the Extinction and the asymmetry parameters making the results valid only in dry sky conditions. This is valid along the paper only in Clean Period (CP, Fig. 2g) but not in half of the profiles measured during Transition Period (TP, Fig. 2h) and Heavy pollution Period (HP, Fig. 2i) when RH reached values up to 90%. Thus, I recommend to maintain the obtained results as baseline but also to add new calculation in supplementary material trying to use the best hygroscopic growth function available for North China Plain and to discuss and compare the related uncertainty (or the range in the radiative transfer calculation) both in optical properties as well as in heating rate profiles.

3- Section 2, Line 93-94: "The in-cloud data in this study was screened out according to in-situ measured RH and liquid water content, thus only the out-of-cloud data is reported here". This comment is related to number 2. If in cloud data were collected and after removed how did you closed the missing parts of profiles to perform radiative transfer calculations. Please add a detaile explanations reporting the frequency of clouds, their altitude and thickness and how you solved the aformentioned issue.

4- Section 2.1 lines 132-133: "All the data related to volume concentration was (better were) corrected for standard temperature and pressure (STP, 1013.25hpa , 273.15K)". The sentence is not clear: did you report the data in STP, or did you transform STP data collected by devices into ambient concentrations at ambient T and P? In the first case I remember you that the feedbacks related to heating rate profiles depends on the ambient values of them and not on the values standardized at STP. Please clarify this point.

5- Section 2.3, DISORT calculations: This section requires a big expansion due to the unreported conditions for those calculation. DISORT calculations were performed in clar-sky approximation? Please clarify and discuss the uncartainties with respect to question 3 due to the presence of clouds during the campaign. At which time DISORT calculations were performed? Noon? With which Zenith angle? Actinic fluxes were calculated and divided in Figure 6 into direct, diffuse up and diffuse down. Connected to this: calculations were perfomed as difference in the model with and without aerosol? Or these data refers to aerosol presence together with standard gaseous atmosphere? How did you close the gap between 2500 m (max altitude of profiles) and the top-of-atmosphere in DISORT application for what concern the aerosol properties? Please specify it clearly.

6- Eq. 4 please cite the reference for this equation.

7- Sections 3.2-3.3-3.4-3.5: despite the issues posed in the previous questions, the ambient discussion reported here is very well described. As these are not the only heating rate BC and BrC data and heating rate profiles available in literature, I strongly suggest you to cite and compare your results with literature data collected in other places of the world to give to your paper a wider view. In this respect your results are increadibly close to those reported in ACP by Ferrero et al. (2014; Atmos. Chem. Phys., 14, 9641–9664) but a comparison is also called for with the good works of Ramana et al. (2010; Nat. Geosci., 3, 542–545, doi:10.1038/ngeo918) and Chakrabarty et al. (2012; Geophys. Res. Lett., 39, L09804, doi:10.1029/2012GL051148) and Kedia et al. (2010; J. Geophys. Res., 115, D07205)

8- Section 3.5 and 3.6 and Figure 8: this part are very important for their implications. however due to the uncertaintes related in TP and HP calculations due to the untreated humidity effect in optical and radiative transfer data, could you compare and discuss the BrC contribution with respect to this point and with other available data? For example experimental BrC heating rate data are available in Ferrero et al. (2018; Environ. Sci. Technol. 52, 3546–3555) while other important data are reported in Chung et al. (2012;

Proc. Natl. Acad. Sci. U.S.A., 109 (29), 11624−11629) and in Shamjad et al. (2015; Environ. Sci. Technol., 49 (17), 10474−10481).

Minor comments are reported here below:

1- Line 266. Fig 7d-f: maybe figure 5d-f
* * *

---

## Author Comment (AC1) · 10 Dec 2019

Response to Referee #2

We thank the Referee for the important comments, which have helped improve the manuscript. The Referee's suggestions are shown in italic font marked as R# and our detailed response/revisions are indexed as A#.

R1. *GENERAL COMMENTS: This manuscript presents aerosol optical properties and heating rate along vertical profiles. Due to the several feedbacks triggered by the vertical forcing behavior I encourage the publication of this work after a serious revision as some parts are not clear enough and other require an improved description. Finally, a comparison with other data available in literature is required. A final improvement of the english is mandatory due to the several typos present along the paper.*

A1. In the revised version, according to the referee's suggestions, the main revisions we have performed including:

1) Incorporate the aerosol hygroscopic effect on the vertical distribution of aerosol optical properties ($\sigma_{sca}$, $\sigma_{ext}$, *SSA*, and *g*), and reanalyze the AOD influenced by aerosol hygroscopic growth.

2) recalculate the radiative transfer by considering the hygroscopic effect.

3) the related discussions are added

4) more comparisons with other literatures are added

5) The English has been improved by native speaker and the typos have been corrected.

*Major comments are reported here below:*

*R.2 1- Introduction and line 75-78: It is obviously clear that the focus of the paper is to describe the situation over Beijing, however a world-wide of existing heating rate data (along vertical profiles as well as at ground) is mandatory concerning the importance of this topic.*

A.2 We have extended our discussions on the related topic for the other regions over the world in the revised manuscript.

Page 3. Line 69. "The light-absorbing aerosol mainly includes the species of black carbon (Bond et al., 2013), brown carbon (Lack and Cappa, 2010) and dust (Klingmüller et al., 2019), which have different spectral sensitivities to solar radiation. Different aerosol components dominate at different environments, and the heating rate caused by various aerosol sources has been studied over the world, e.g. for the anthropogenic sources over north America (Gao et al., 2008; Sahu et al., 2012; Liu et al., 2015b), Europe (Ferrero et al., 2014; Ferrero et al., 2018) and south Asia (Chakrabarty et al., 2012; Shamjad et al., 2015), and biomass burning sources over north and south America (Saleh et al., 2015; Zhang et al., 2017). However the data is still sparse regarding the vertical structures of heating rate, in addition to that, the heating was mostly evaluated using the measurement on the surface (Mallet et al., 2008; Wang et al., 2009) rather than using directly measured vertical profile. The calculation was performed for single species such as BC or BrC but most did not consider the co-impacts of all species (Chakrabarty et al., 2012; Chung et al., 2012; Shamjad et al.,

2015). In the lower free troposphere, the heating rate of aerosol in interacting with boundary layer dynamics has raised much attention recently, as it may play important role in depressing boundary layer development hereby exacerbating the local pollution (Li et al., 2017). The heating rate caused by light absorbing aerosol was reported to vary as a function of height and range at 0.3-2.1 K/day for the polluted PBL over Europe (Kedia et al., 2010; Ferrero et al., 2014; Ferrero et al., 2018), and 0.3-2.5 K/day for south Asia (Tripathi et al., 2007; Ramana et al., 2007; Ramachandran and Kedia, 2010; Chakrabarty et al., 2012), but limited reports for the region of polluted east Asia."

*R.3 2- Section 2, Line 88 and section 3.4 lines 263-265: "silicate direr (change to dryer)instruments was utilized (better used) ahead all instruments to maintain the sampling RH lower than 40%" and "Improved agreement between both may be achieved by considering the particles hygroscopic growth, which requires composition measurement to constrain this factor but this was not available in this study". This is a serious lack in the work due to the aim to perform radiative transfer calculations. The choice to measure dry aerosol optical properties (especially for scattering) seriously affect the SSA, the Extinction and the asymmetry parameters making the results valid only in dry sky conditions. This is valid along the paper only in Clean Period (CP, Fig. 2g) but not in half of the profiles measured during Transition Period (TP, Fig. 2h) and Heavy pollution Period (HP, Fig. 2i) when RH reached values up to 90%. Thus, I recommend to maintain the obtained results as baseline but also to add new calculation in supplementary material trying to use the best hygroscopic growth function available for North China Plain and to discuss and compare the related uncertainty (or the range in the radiative transfer calculation) both in optical properties as well as in heating rate profiles.*

A.3 In the revised manuscript, the hygroscopic growth of aerosol is estimated by the previous $f(\text{RH})$ measurements conducted over the same region. This gives the enhancement of particle scattering coefficient as a function of ambient RH, we have adopted this function to our data for a best estimate on the potential water growth influence on the aerosol scattering coefficient, aerosol extinction coefficient, single scattering albedo and asymmetry parameter, hereby the in-situ measured and remote sensing AOD are further compared, and the new profiles are input to the radiative transfer model to work out the updated actinic flux. The scattering and extinction enhancement is only appreciable for ambient RH >40% which applied for four of our flights. This excludes three of the flights when the boundary layer clouds were experienced and the AERONET products were not available, thus the comparisons were not performed for these flights.

At Page 7. Line 167. the calculation we performed:

"To evaluate the potential influence of particle hygroscopicity on optical properties, the aerosol hygroscopic growth parameterization ($f(\text{RH})$) was used to calculate the enhancement of $\sigma_{sca}$ under ambient RH. This function was previously measured by Zhao et al. (2019b) over Beijing region, expressed as:

$$f(RH) = a \cdot (1 - RH/100)^{-\gamma(RH/100)}$$

(4)

where $f$(RH) was obtained by a comparison between a dry and humidified nephelometer in parallel. a / γ was 0.930 / 0.329, 0.971 / 0.372, and 0.988 / 0.356 for clean, moderate, and heavy pollution period, respectively, according to the study.

The RH influence on $g$ was calculated according to Zhao et al. (2018), expressed as:

$$g(RH)/g(RH < 40\%) = a \cdot (1 - RH/100)^{-\gamma(RH/100)}$$
(5)

where a / γ was 0.9984 and 0.0849.

The resulting $\sigma_{sca}$, $\sigma_{ext}$, $SSA$, and $g$ are all calculated for the hygroscopicity influence."

Then we performed further comparison with AERONET using aerosol hygroscopicity-corrected AOD. The updated profiles of aerosol optical properties are also input in radiative transfer calculation to evaluate the updated heating rates.

Page 11. Line 268. "The hygroscopic effect on aerosol vertical profiles was mainly controlled by the ambient RH (shown in blue lines in Fig. 3). For most of the flights, the hygroscopic effect could be neglected due to low RH (< 50%) (Fig. 2). For some of the flights (20161211), $\sigma_{sca}$ and $\sigma_{ext}$ especially at top of the PBL could be enhanced by a factor of 1.3."

Page 12. Line 294. "Note that only one flight (flight 20161211) under RH > 80 %, the particle hygroscopicity had appreciable influence on $SSA$ (increased by 0.05), $SAE$ (decreased by 0.2) and $g$ (increased by 0.1)."

The related discussions are added:

Page 13. Line 320. "Improved agreement was achieved by 8-15% if considering aerosol hygroscopic growth (open circle in Fig. 5a-c), despite that in-situ constrained AOD was still 2-5% lower than AERONET after the hygroscopic correction."

Page 14. Line 351. "The AF received at lower level was reduced by up to 10 % by incorporating the aerosol hygroscopicity influence (Fig. 7) due to enhanced AOD, and AF was further redistributed to give larger vertical gradient (Fig. 7a-c). "

Page 15. Line 375. "Corresponding with the aerosol hygroscopicity influence on the actinic flux, the heating rate showed lowered intensity but enhanced vertical gradient for the flights with high ambient RH (Fig. 7b)."

Fig. 3, Fig. 4, Fig. 5, Fig.6 and Fig. 7 are also updated to include the aerosol hygroscopic effect.

*R. 4: Section 2.1 lines 132-133: "All the data related to volume concentration was (better were) corrected for standard temperature and pressure (STP, 1013.25hpa ,*

*273.15K)".The sentence is not clear: did you report the data in STP, or did you transform STP data collected by devices into ambient concentrations at ambient T and P? In the first case I remember you that the feedbacks related to heating rate profiles depends on the ambient values of them and not on the values standardized at STP. Please clarify this point.*

A.4. This point has been clarified in the revised version:

Page 7. Line 156. "The parameters $\sigma_{sca}$, $\sigma_{abs}$, and $\sigma_{ext}$ are reported as standard temperature and pressure (STP, 1013.25hpa , 273.15K) for direct comparison at different altitudes among flights. Note that to compare with the AERONET results and for the radiative transfer calculation (as detailed in the following), these parameters in ambient conditions are used. "

*R. DISORT calculations were performed in clear-sky approximation? Please clarify and discuss the uncertainties with respect to question 3 due to the presence of clouds during the campaign.*

A. 5 We have added the related description in the revised version. In addition to that, for clarification, we have moved the previous Table S1 which in detail describes the input parameters used in DISTORT to the main texts as Table 2. The calculation is performed for clear-sky condition only, thus the flights experiencing low-level clouds are not included in the calculation."

Page 5. Line 109. "In order to compare the AOD from AERONET and calculate the vertical heating rates, only the cloud-free vertical profiles are used. In this study, three flights (20161117 12:00, 20161117 15:00, 20161118 12:00) were observed with cumulus clouds (Table 1). The in-cloud data in this study was screened out according to the in-situ measured cloud number concentration and liquid water content, with a total number concentration of more than 10 $cm^{-3}$ and liquid water of more than 0.001 g $m^{-3}$ are not included in the following analysis (Deng et al., 2009)."

Page 8. Line 183. "The aircraft in-situ measured vertical profiles of AOD, single scattering albedo ($SSA$) and $g$ are used as inputs, and the other input parameters for the radiative transfer calculation is summarized in Table 2. The calculation is performed for clear-sky condition only, thus the flights experiencing low-level clouds are not included in the calculation."

*At which time DISORT calculations were performed? Noon? With which Zenith angle?*
Reply: The time of each specific profile was used in the DISORT calculation, as stated in the Table 2, and the Zenith angle was calculated based on the location and time for each flight.

*Actinic fluxes were calculated and divided in Figure 6 into direct, diffuse up and*

*diffuse down. Connected to this: calculations were performed as difference in the model with and without aerosol? Or these data refers to aerosol presence together with standard gaseous atmosphere?*

Reply: The calculation is performed with and without aerosol input (AOD is set to zero) to evaluate the aerosol impact. The heating rate is only calculated with considering the in-situ measured AOD. The gas uses the standard gaseous atmosphere as stated in Table 2.

This is added in the revised version:

Page 8. Line 187: "The calculation of AF is performed with and without aerosol input (AOD is set to zero) to evaluate the aerosol net impact. The heating rate is only calculated with considering the in-situ measured AOD."

*How did you close the gap between 2500 m (max altitude of profiles) and the top-of atmosphere in DISORT application for what concern the aerosol properties? Please specify it clearly.*

This information has been added.

Page 7. Line 162: "The measurement of $\sigma_{ext}$ was up to 2500m above which the aerosol concentration was low enough to be below the instrument lower detection limit. Given the very low concentration above 2500m, the value on top of 2500m was used to reconstruct the vertical profile up to 5000m. After that the $\sigma_{ext}$ from 2.5-5 km only accounted for 1-2 % of the integrated columnar extinction."

*R6. Eq. 4 please cite the reference for this equation*
A6. Reply: added.

*R7. Sections 3.2-3.3-3.4-3.5: despite the issues posed in the previous questions, the ambient discussion reported here is very well described. As these are not the only heating rate BC and BrC data and heating rate profiles available in literature, I strongly suggest you to cite and compare your results with literature data collected in other places of the world to give to your paper a wider view. In this respect your results are incredibly close to those reported in ACP by Ferrero et al. (2014; Atmos. Chem. Phys., 14, 9641–9664) but a comparison is also called for with the good works of Ramanaet al. (2010; Nat. Geosci., 3, 542–545, doi:10.1038/ngeo918) and Chakrabarty et al. (2012; Geophys. Res. Lett., 39, L09804, doi:10.1029/2012GL051148) and Kediaet al. (2010; J. Geophys. Res., 115, D07205).*

*R8. Section 3.5 and 3.6 and Figure 8: this part are very important for their implications. however due to the uncertainties related in TP and HP calculations due to the untreated humidity effect in optical and radiative transfer data, could you compare and discuss the BrC contribution with respect to this point and with other available data? For example experimental BrC heating rate data are available in Ferrero et al. (2018; Environ. Sci. Technol. 52, 3546–3555) while other important data are reported in Chung et al. (2012; Proc. Natl. Acad. Sci. U.S.A., 109 (29),*

*11624-11629) and in Shamjad et al. (2015; Environ. Sci. Technol., 49 (17), 10474-10481).*

Reply: We have cited and discuss the references the referee mentioned in the introduction and discussion.

Page 15. Line 362. "The results here show that the atmospheric heating by aerosol was mainly inside the PBL and for polluted period the BC-induced heating was 0.05-0.17 K/h, generally consistent with previous studies over the polluted Asia region, with 0.02-0.17 K/h (Ramana et al., 2007; Ramana et al., 2010; Kedia et al., 2010)."

Page 15. Line 372. "The contribution of BrC to the total absorption was reported to be 10-27 % over polluted region of Europe (Ferrero et al., 2018) and south Asia (Chung et al., 2012; Shamjad et al., 2015), in general consistent with results during polluted periods here."

Page 15. Line 383. "This study showed positive vertical gradient for 30 % of the flights especially under high pollution, and in particular during regional transport when pollutants were advected from outside of Beijing and showed elevation of absorption at higher altitude (Fig. 8). The rest of the flights showed highly accumulated aerosol concentration near surface, also found by a previous study (Ferrero et al., 2014), when BC wound potentially promoted the dispersion in the PBL and decreased its stability."

---

## Author Comment (AC2) · 10 Dec 2019

Response to Referee #1

We thank the Referee for the comments, which have helped improve the manuscript. The Referee suggestions are shown in italic font marked as R# and our detailed response/revisions are indexed as A#.

R1. *GENERAL COMMENTS:*

*This manuscript presents a valuable observational dataset of in-situ aircraft measurement of BrC and BC optical profiles in Beijing. The corresponding influences on heating rate and radiative forcing are analyzed and extensively compared with AERONET dataset. Although the pollution and meteorology interaction over North China Plain (NCP) were widely investigated through surface observations and modelling, limited studies have considered the evolution of pollutants in vertical profile. This work could fill this gap well. The method and uncertainties are well described and discussed. The manuscript is well-written, but some parts of it are not clear enough. I would recommend for publication after the authors address the following specific comments:*

A1. We thank the positive comments from the referee.

*SPECIFIC COMMENTS*

*R1. line 22: "replying" or "relying" ?*

A1. This typo is corrected.

*R2. line 68-69: "and regional transport will introduce enhanced aerosol loading to high level". Please introduce the corresponding vertical transport processes in more details base on previous studies over NCP, such as the influence of mountain valley breeze led by the special topography over Beijing.*

A2. We have added the mountain chimney effect in the revised manuscript.

P3, Line 68: "such as the mountain chimney effect over Beijing region may introduce enhanced aerosol loading to high level (Chen et al., 2009)."

**R3. line 73. Suggest delete "successive", which is too subjective.**

A3. Corrected.

**R4. line 77. Suggest delete "for the first time". There are lots of previous studies regarding aerosol optical property observations over NCP, although may not elaborate the detailed BrC properties as this work does. The "first time" description is not appropriate here.**

A4. It has been corrected in the revised manuscript.

**R5. line 86. At which temperature level are maintained?**

A5. The information is added:

Page 5. Line 103. "The maintained room temperature (25 $^{\circ}$C) in the cabin had selfdrying effect when the temperature inside was higher than outside the cabin, in addition to which, a silicate direr was utilized ahead of all instruments to maintain the sampling RH lower than 40%."

**R6. line 93. Please give the criteria of screening out the "in-cloud data".**
A6. We have added the following:
Page 5. Line 112. "The in-cloud data in this study was screened out according to the in-situ measured cloud number concentration and liquid water content, with a total number concentration of more than 10 $cm^{-3}$ and liquid water of more than 0.001 g $m^{-3}$ are not included in the following analysis (Deng et al., 2009)."

**R7. line. 116. "which is independent of the filter artifacts". I do not understand here.**
A7. This has been clarified as:
Page 6. Line 138. "The multiple scattering artifact of AE33 was corrected by measuring the ambient aerosol in parallel with photoacoustic spectrometer (PASS3, DMT Inc, USA), and latter is independent of the filter artifacts."

**R8. Eq.3. Please modify it as the same format of the Eq.1.**
A8. Corrected.

**R9. line 152. Please specify the values (with units) of "air mass density" and "Cp" here.**
A9. Corrected.
Page 9. Line 197: "where $\rho$ and $C_P$ are the air mass density ($kg/m^3$) and heat capacity (1.007 J/g*K), respectively."

**R10. line 269. Where was the study of Andrews et al. (2017) conducted?**
A10. This has been added:

Page 13, Line 326: "This is consistent with previous findings conducted over US that the retrieved AAOD from AERONET was biased higher when compared to in-situ measurement (Andrews et al., 2017)."

**R11. line 384. They are the heating rate at which level? And are they the rate at noon time? Please clarify it.**
A11. This is added:
Page 18. Line 454: "BC was the main heating species, having 0.05 K/h, 0.1 K/h and 0.15 K/h heating rate at local time 12:00 to 15:00 in the PBL during pollution initialization, transition and full development respectively,"

**R12. line 386. "when regional transport". I do not understand here.**
A12. this is revised:
Page 18. Line 456: "showed positive vertical gradient of heating during regional

transport period when pollution was advected at high level from the polluted south region outside of Beijing (Tian et al., 2019)."

***R13. line 386. "contribution of BrC" to what? You mean contribution to the aerosol mass or heat rate or light absorption, or what?***
A13. This is revised.
Page 18. Line 457: "The contribution of BrC to heating rate was found to increase by 20 % throughout the column from CP to HP period"

***R14. Figure 1. The title of Fig. 1c is difficult to understand. And please add labels for the colorbar.***
A14. The labels are added for the colorbar now.

***R15. Figure 2 and following profile figures. Here, use blue, black and red to indicate the clean, transition and polluted period, respectively. However, it is ambiguous that in one profile belongs to two different period (black for lower part, but red for upper part). In my understanding or the common understanding the "period" is separated by time windows.***
A15. In this figure, vertical profiles for clean, transition and heavy pollution period was shown in the left, middle and right panel. Black and red color in each panel was used to denote inside and above the PBL.

***R16. Figure 4. The quality of this figure is poor and unreadable.***
A16. This figure is revised.

---

## Author Response (AR2)

Dear Editor,

Thanks for your detailed suggestions on improving the English of this manuscript, and we have carefully gone through the English (by ourselves and an English native speaker). We believe the English is now meeting with the requirements.

Yours Sincerely,

Ping Tian